# Changes in Gut Bacterial Translation Occur before Symptom Onset and Dysbiosis in Dextran Sodium Sulfate-Induced Murine Colitis

M. Taguer,[a] E. Darbinian,[a] K. Wark,[a] A. Ter-Cheam,[b] D. A. Stephens,[b] C. F. Maurice[a]

[a]Department of Microbiology & Immunology, Faculty of Medicine and Health Sciences, McGill University, Montreal, Quebec, Canada
[b]Department of Mathematics and Statistics, Faculty of Science, McGill University, Montreal, Quebec, Canada

**ABSTRACT** Longitudinal studies on the gut microbiome that follow the effect of a perturbation are critical in understanding the microbiome's response and succession to disease. Here, we use a dextran sodium sulfate (DSS) mouse model of colitis as a tractable perturbation to study how gut bacteria change their physiology over the course of a perturbation. Using single-cell methods such as flow cytometry, bioorthogonal noncanonical amino acid tagging (BONCAT), and population-based cell sorting combined with 16S rRNA sequencing, we determine the diversity of physiologically distinct fractions of the gut microbiota and how they respond to a controlled perturbation. The physiological markers of bacterial activity studied here include relative nucleic acid content, membrane damage, and protein production. There is a distinct and reproducible succession in bacterial physiology, with an increase in bacteria with membrane damage and diversity changes in the translationally active fraction, both, critically, occurring before symptom onset. Large increases in the relative abundance of *Akkermansia* were seen in all physiological fractions, most notably in the translationally active bacteria. Performing these analyses within a detailed, longitudinal framework determines which bacteria change their physiology early on, focusing therapeutic efforts in the future to predict or even mitigate relapse in diseases like inflammatory bowel diseases.

**IMPORTANCE** Most studies on the gut microbiome focus on the composition of this community and how it changes in disease. However, how the community transitions from a healthy state to one associated with disease is currently unknown. Additionally, common diversity metrics do not provide functional information on bacterial activity. We begin to address these two unknowns by following bacterial activity over the course of disease progression, using a tractable mouse model of colitis. We find reproducible changes in gut bacterial physiology that occur before symptom onset, with increases in the proportion of bacteria with membrane damage, and changes in community composition of the translationally active bacteria. Our data provide a framework to identify possible windows of intervention and which bacteria to target in microbiome-based therapeutics.

**KEYWORDS** bacterial physiology, high nucleic acid, low nucleic acid, flow cytometry, colitis, BONCAT, longitudinal

The diversity of the gut microbiota has been characterized in a range of settings, leading to an increased understanding of the critical and complex roles that it has in host health (1–7). While the mechanisms underlying the role of the gut microbiome in disease are beginning to be unraveled, developing microbiome-targeted therapeutics remains challenging. This may be due in part to the wealth of cross-sectional studies that focus on determining significant bacterial compositional changes and the search for specific taxa associated with disease. Indeed, cross-sectional studies of

Address correspondence to C. F. Maurice, corinne.maurice@mcgill.ca.

The authors declare no conflict of interest.

disease-associated gut microbiomes do not provide information on the microbial succession underlying disease progression, a requisite for successful microbial intervention. Based on the steady-state framework of microbiomes, once the microbiome is in a steady state, whether homeostasis or an altered composition associated with disease, the microbial community has a high level of resilience (8–10). This makes it very difficult to introduce sustained changes in the microbial community (see Sommer et al. [8] for an excellent review).

Longitudinal studies are essential to identify time frames when the microbiome is not yet in steady state and thus most amenable to modulation. A reliable and reproducible perturbation model is also critical for characterizing gut microbial succession to disease. The dextran sodium sulfate (DSS) model of colitis is well established in mice, with various prognoses based on genetic background (11, 12). C57BL/6 mice exhibit signs of intestinal colitis, such as intestinal inflammation, a shortened colon with mucosal damage, and rectal bleeding, typically after 3 to 7 days of DSS administration in their drinking water. The gut microbiome is heavily implicated in colitis: its presence is required for a robust colitis, its transfer can induce colitis, and its composition and diversity are altered during the inflammatory period (13–15). After DSS cessation, mice recover from the intestinal damage, inflammatory markers decrease (12), and the gut microbiota returns to baseline diversity levels, although not fully (16). The changes in bacterial community composition are consistent, with decreases in the short-chain fatty acid (SCFA) producers *Clostridiales* and increases in the more oxygen-tolerant and proinflammatory *Enterobacteriaceae* (16, 17). These DSS colitis-specific and consistent changes can be considered a dysbiosis, defined as when "the microbiota crucially contributes to the manifestation or continuation of a given disease that cannot be attributed to a single bacterial species" (8). This dysbiosis has been suggested to be an alternative steady state of the microbiome (9), which correlates with disease severity and inflammation (16–19). Given the difficulties in inducing long-term changes in a microbiome at steady state, it is crucial to characterize the functional succession to dysbiosis to identify bacterial targets for modulation. However, the changes undergone by microbial communities between these distinct steady states (homeostasis and dysbiosis) remain poorly described.

Recent efforts have aimed at characterizing the longitudinal dynamics of the gut microbiome in colitis, with a focus on its links to disease severity, remission outcome, and treatment response; yet these studies have been unable to find biomarkers of relapse or remission (20–25). All but one (21) of these studies focused on bacterial community composition and potential metabolism through metagenomics, as opposed to actual microbial functionality through metatranscriptomics or metabolomics. To characterize the dynamics of the gut microbiota and its succession to dysbiosis, a focus on bacterial activity, rather than community composition, is needed. Indeed, changes in bacterial functionality are not always reflected in changes in diversity or metabolic pathways, as bacteria can modulate their activity through transcriptional, translational, and posttranslational modifications typically missed in DNA-focused approaches (26–30). Certain 'omics techniques, such as metatranscriptomics, provide functional information on bacterial communities, but they remain limited by incomplete databases and often cannot link a given function to specific taxa. In addition, next-generation sequencing loses quantitative information, transforming diversity into a compositional framework without information on clonal differences (31, 32). The inability to link bacterial activity to diversity limits our understanding of bacterial interactions in complex communities and their response to perturbations. Determining the longitudinal functional changes of bacterial communities between alternative stable states and linking them to specific bacterial taxa would provide a causative framework for targeted interventions focusing on activity and community resilience, which is currently lacking.

Single-cell methods can rapidly isolate and identify physiologically distinct bacteria from communities, potentially allowing us to link physiology to taxonomic identity, without cultivation (27, 33). Fluorescence-activated cell sorting and subsequent sequencing (FACS-Seq)

is able to rapidly discriminate bacteria based on optical characteristics, such as size, shape, intracellular density, and fluorescent properties of various physiological dyes (34–37). Here, we seek to determine the microbial functional succession to dysbiosis in a DSS mouse model of colitis by following three distinct markers of bacterial activity at a fine temporal scale: (i) nucleic acid content, (ii) membrane damage, and (iii) protein production through bioorthogonal noncanonical amino acid tagging (BONCAT). These markers encompass broad, yet distinct, physiological traits that are closely linked to bacterial metabolic activity (35, 38–41). When stained with nucleic acid dyes that stain both DNA and RNA (42), bacteria cluster into two main cytometric populations according to their nucleic acid content and resulting levels of fluorescence. Multiple studies suggest that the highly fluorescent bacteria are metabolically more active than their less fluorescent counterparts (33, 39, 43–45), supporting the use of this broad physiological marker of metabolic activity. Membrane damage can be identified through membrane exclusion dyes, which stain damaged or dead bacteria (46), while BONCAT detects translationally active cells through the incorporation of noncanonical amino acids and fluorescent labeling with click chemistry (41). This allows for unbiased, single-cell resolution of translationally active bacteria while they are still in their complex community. By sorting and sequencing these cytometric populations multiple times along the progression of DSS-induced colitis, we quantitatively and qualitatively monitored the microbial succession to dysbiosis. Our data show a distinct reproducible physiological succession to a dysbiosis typically associated with colitis, led first and foremost by the translationally active bacteria. These microbial functional alterations occurred prior to the development of inflammatory symptoms and progressed with increased relative abundances of *Akkermansia muciniphila*. Our work provides insight into the dynamics of bacterial interactions during alternative steady states, providing more sensitive information than diversity metrics alone. Understanding how bacteria change their physiology and activity in response to perturbations may elucidate a critical window during which microbiome-targeted therapeutics would be most effective.

## RESULTS

**DSS-induced colitis causes transient changes in the proportions of physiologically distinct bacteria.** Two cages of five C57BL/6 male mice each (cages A and C) and one cage of five female mice (cage B) were independently exposed to 2% DSS in drinking water for 5 days to induce colitis (Fig. 1A). Fecal samples were collected prior to (2 to 4 samples), during (5 samples), and after (5 samples) colitis, until the mice had no more blood in stool. The onset of colitis was determined for each mouse on each sampling day with the disease activity index (DAI), encompassing weight loss, stool consistency, and the presence of blood in stools (Fig. 1B; see also Fig. S1 in the supplemental material), and these results were complemented by determining increases in fecal lipocalin 2 levels (Fig. 1C). Based on these results, the longitudinal study was broken up into four disease states: baseline, presymptomatic, symptomatic, and recovery. Baseline corresponds to the days before DSS administration (days −3 to 0), the presymptomatic state is when the mice are exposed to DSS but have minimal symptoms (DAI < 5), the symptomatic state is when the mice have a DAI of >5 and increased lipocalin 2 levels, and recovery days are identified when the mice have no more blood in their stool (DAI < 5). After recovery, there are two weekly follow-up sampling days (W1 and W2).

Three aspects of bacterial physiology were monitored: (i) relative nucleic acid content, (ii) membrane damage, and (iii) protein production. Relative nucleic acid content was assessed through two fractions discriminated by flow cytometry: bacteria with a high nucleic acid content (HNA) and bacteria with a low nucleic acid content (LNA). Within a given system, the HNA bacteria have higher levels of metabolic activity than their LNA counterparts (39, 44). Membrane damage is monitored through staining with propidium iodide (PI), a membrane exclusion dye, and protein production is monitored through BONCAT labeling.

Total cell counts and the proportions of cells in each physiological fraction were determined through flow cytometry, and the data represent means of results from all 3 cages to limit cage effects and highlight the reproducibility of results. While there were

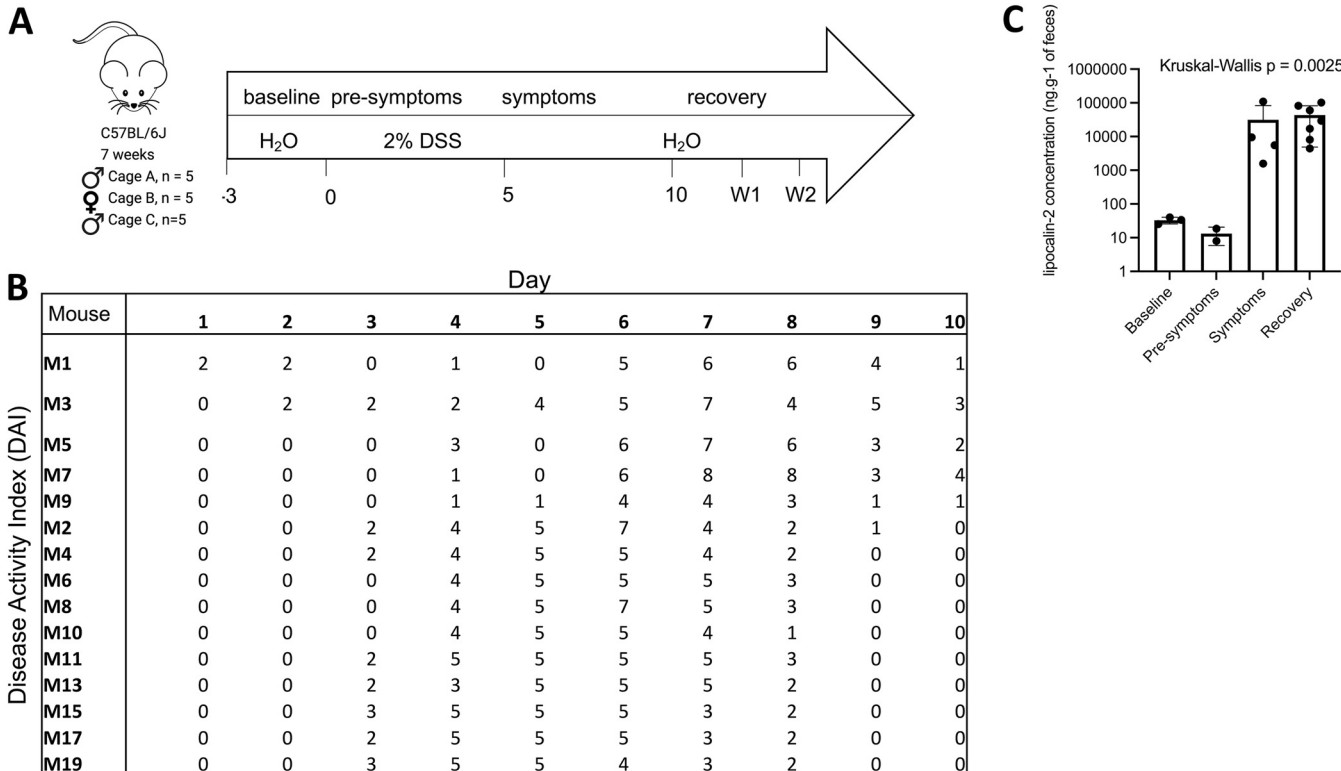

**FIG 1** Two percent DSS consistently and reproducibly induces colitis. (A) Sampling timeline with disease states identified; (B) disease activity index for each mouse on each sampling day; (C) fecal lipocalin 2 concentrations in cage C only. Error bars represent standard deviations.

no significant changes in bacterial load over time or between cages ($P > 0.1$) (Fig. 2A), the proportions of each physiological fraction changed before or during symptom onset. There was a significant increase in the proportion of bacteria with membrane damage during the presymptomatic phase compared to that at baseline, from 11.8 $\pm$ 7.4% to 15.2 $\pm$ 8.6% ($q = 0.014$) (Fig. 2B). There was a decrease in the proportion of translationally active bacteria during the presymptomatic phase, from 76% $\pm$ 14% to 61% $\pm$ 24% (Fig. 2C). The proportion of HNA bacteria decreased during DSS administration from 49.3 $\pm$ 15.8% at baseline to 43.7 $\pm$ 17.5% at the peak of symptoms (slope $m = -5.07$; $P < 0.05$). The proportion of HNA cells then began to recover to baseline levels ($m = 3.67$; $P = 0.08$) (Fig. 2D, left). As the distinction between HNA and LNA cytometric populations were not always clear during the DSS perturbation, the median fluorescence intensity (MFI) of the entire bacterial community was also calculated. The MFI decreased simultaneously with the proportion of HNA bacteria, representing a total loss in nucleic acid content of the cells between baseline and the presymptomatic state ($m = -1663$; $P < 0.0001$) (Fig. 2D, middle). The RNA-to-DNA ratios measured by fluorometric quantification follow similar trends, albeit once the mice are symptomatic, with the largest drop in the RNA/DNA ratio occurring at peak symptom severity (Fig. 2D, right).

As sex-specific responses have been noted in DSS-induced colitis before, a breakdown by sex is in Fig. S2 (47). While there are differences in how bacterial physiology changes in response to DSS perturbation, we stress that due to the low sample sizes for each sex, we cannot attribute these differences to sex-specific effects; further experiments with larger sample sizes would be needed to confirm the differences seen here. Overall, sex accounted for 4.4% and 8.76% of the variation in the proportion of propidium iodide-positive ($PI^+$) bacteria and BONCAT-labeled bacteria, with no significant effect of sex on the proportion of HNA bacteria over time.

In addition, regressions were performed for each physiological fraction to determine changes over time. Regressions using a Bayesian linear mixed model indicate a reduction in HNA, PI, and BONCAT cells during the symptomatic disease state. Cage and intermouse effects were modest, explaining little of the variation, with time being

mSystems®

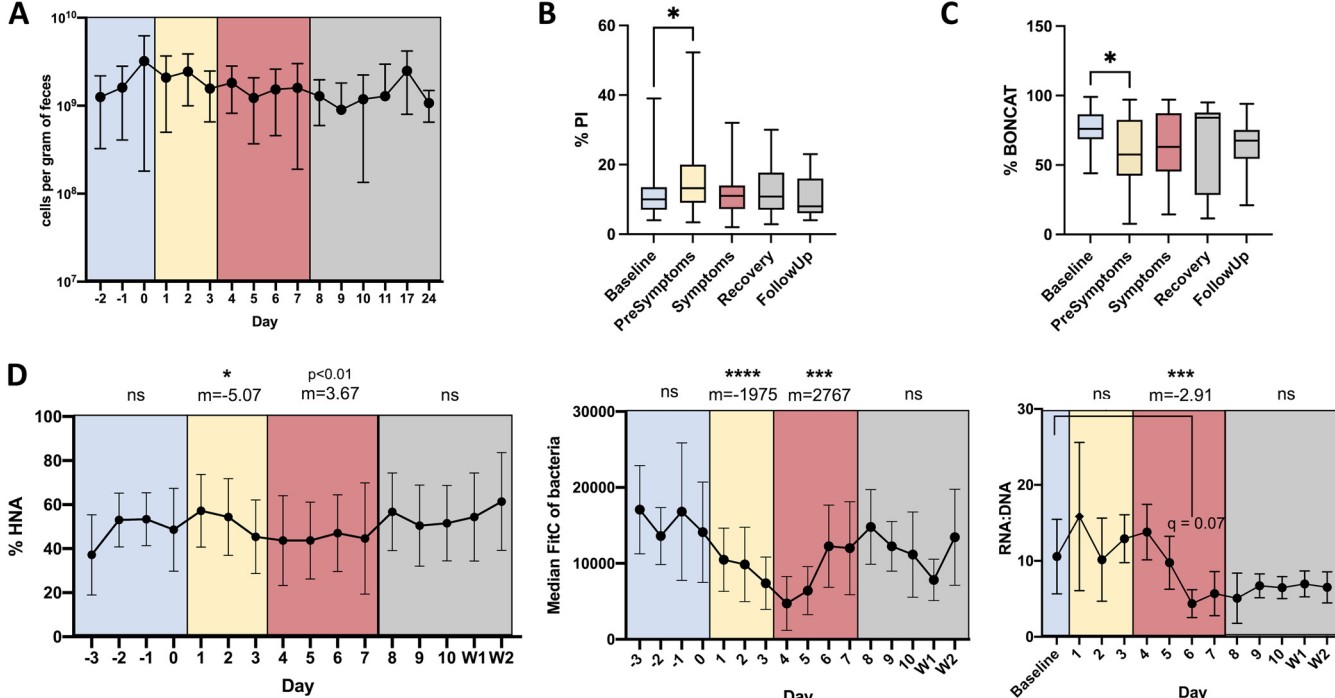

**FIG 2** Proportions of physiologically distinct bacteria change prior to symptom onset. (A) Average bacterial abundance per gram of feces from all mice (n = 15). (B) Proportion of bacteria with membrane damage. (C) Proportion of translationally active bacteria. (D, left) Proportion of HNA bacteria; (center) median fluorescein isothiocyanate (FITC) fluorescence of the entire bacterial population; (right) RNA/DNA ratios of the whole community. The slope for each disease state, as well as the statistical significance, is indicated at the top. (C, center and right) Data are from one representative experiment; all other panels represent the averages of results from 3 experiments (n = 15 total), and error bars represent standard deviations. Color-coding represents the disease state according to DAI and lipocalin 2 assays (blue: baseline, yellow: presymptomatic, red: symptomatic, grey: recovery, with recovery divided into immediate recovery and the two weekly follow-ups [W1 and W2]). Paired, mixed-effects analyses were performed to test for statistical significance against baseline values, correcting for multiple comparisons using the Geisser-Greenhouse correction. ns, not significant; *, $P < 0.05$; **, $P < 0.005$; ***, $P < 0.0005$; ****, $P < 0.0001$.

the largest explanatory variable, again signifying that the succession to disease is distinct and reproducible. Overall, changes in bacterial physiology occur before or in concordance with the onset of symptoms.

**DSS causes consistent changes in bacterial diversity across physiological groups.** All physiological groups of bacteria monitored (HNA, LNA, PI, and BONCAT) were sorted, and the V4-V5 region of the 16S rRNA gene was sequenced to determine the composition changes in each physiological fraction during a perturbation. The ordinations of beta diversity distances between physiological groups show significant clustering based on the disease state for each physiological fraction, except for PI+ bacteria (Fig. 3), indicating that these different physiologies are dynamic in response to DSS-induced colitis. As the PI population is low (mean, 12.4% across all days), PI sorting was pooled by cage and day, lowering the power for PI diversity analyses.

As disease state had a clear effect on the diversity of each physiological fraction, we next wanted to determine how much the communities were changing compared to baseline. Pairwise beta diversity distances were calculated from each disease state to baseline. Similar trends were seen for each physiological fraction, with communities diverging further away from baseline as disease progressed and with diversity starting to return to baseline during recovery (Fig. 4A). Most physiological fractions were still different from baseline even once symptoms had disappeared (weekly follow-ups were not included), suggesting that gut microbial communities remain altered even after host recovery and loss of symptoms. To focus on the succession to colitis and the associated dysbiosis, we analyzed the changes in the presymptomatic disease state to determine which physiological group was changing the most. Comparing the levels of divergence between the presymptomatic state to baseline, the BONCAT fraction changed the most, notably more than the whole community (Fig. 4B). This suggests

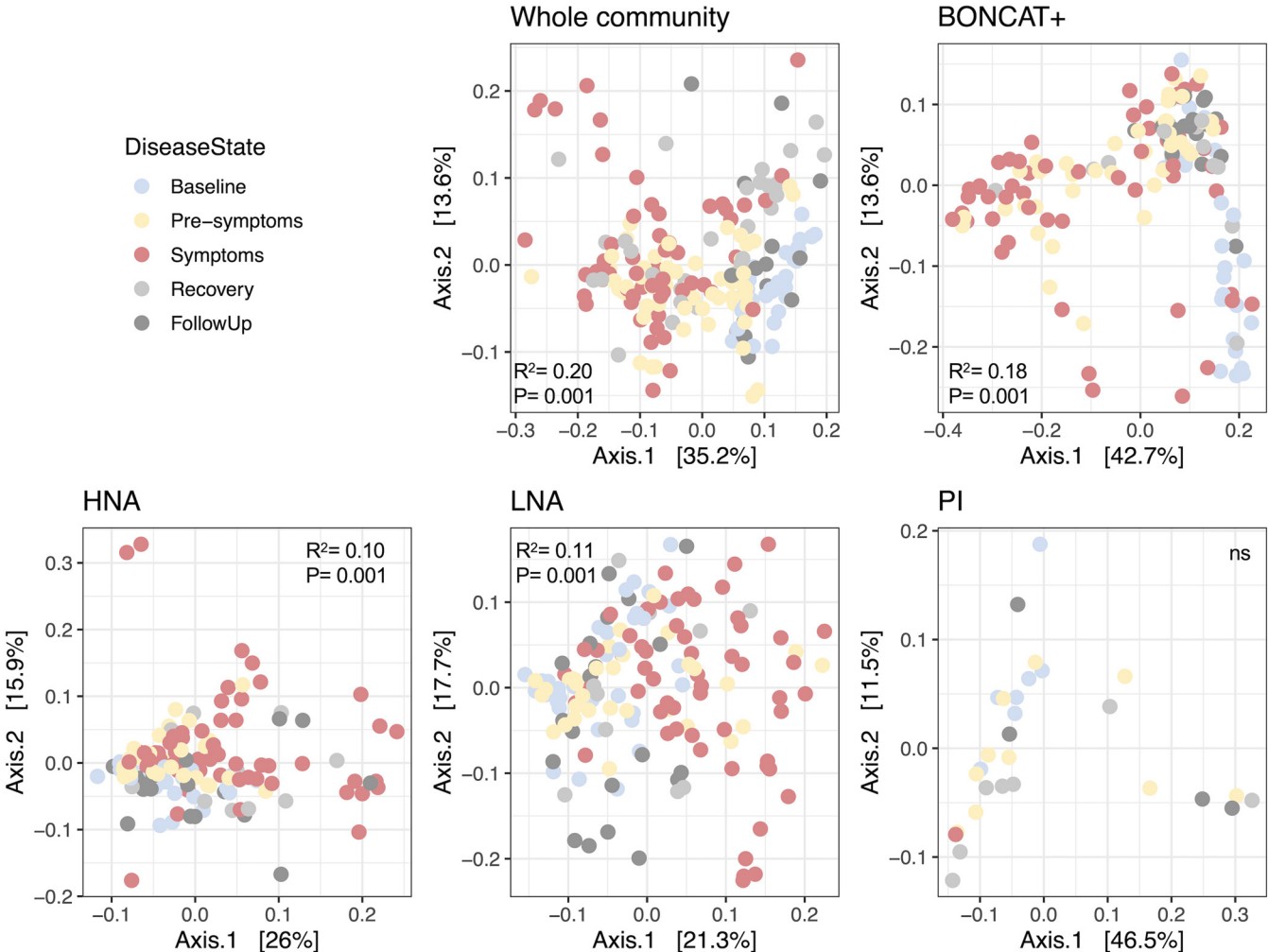

**FIG 3** Disease state has an effect on the beta diversity of physiological fractions. Principal-component analysis (PCoA) of weighted UniFrac distances by physiological fraction. Samples are colored based on disease state. PERMANOVA results of the effect of the disease state on variation are included within each ordination plot. Whole community, unsorted original sample; BONCAT, click-labeled protein-producing bacteria; HNA, high-nucleic-acid-content bacteria; LNA, low-nucleic-acid-content bacteria; PI, propidium iodide-stained bacteria with membrane damage; W1, weekly follow-up 1; W2, weekly follow-up 2.

that while each physiological fraction follows the same trend over time, the earliest changes occur primarily in the translationally active bacteria, and these changes are more pronounced than in the whole community.

**Unique taxa are associated with colitis between physiological groups.** We next wanted to determine which bacteria were changing in abundance across disease states and physiological fractions. While physiological groups are distinct from one another in each disease state ($R^2 = 0.049$; $P < 0.001$) (Table 1), they maintain similar trends in diversity over time. These trends include increases in the numbers of members of the *Bacteroidetes* (*Bacteroides*) and *Verrucomicrobia* (*Akkermansia*) and decreases in *Firmicutes* (*Lachnoclostridium*, *Lachnospiraceae*, *Dubosiella*, *Turicibacter*) (Fig. 5). The bacterial community remained stable during baseline at the phylum, genus, and amplicon sequence variant (ASV) levels ($P > 0.1$, by permutational multivariate analysis of variance [PERMANOVA] of physiology to day) and began to change within 48 h of DSS administrations. Peak changes in community structure were concurrent with peak inflammation, with a return to baseline levels by the end of the experiment (Fig. 5). Across physiological groups and disease states, all major phyla are differentially abundant (*Actinobacteria*, *Bacteroidetes*, *Firmicutes*, *Proteobacteria*, *Tenericutes*, and *Verrucomicrobia*; by analysis of composition of microbiomes II [ANCOM II], W statistic = 5, cut-off = 0.9). At the genus level, 60 genera out of 67 were considered differentially

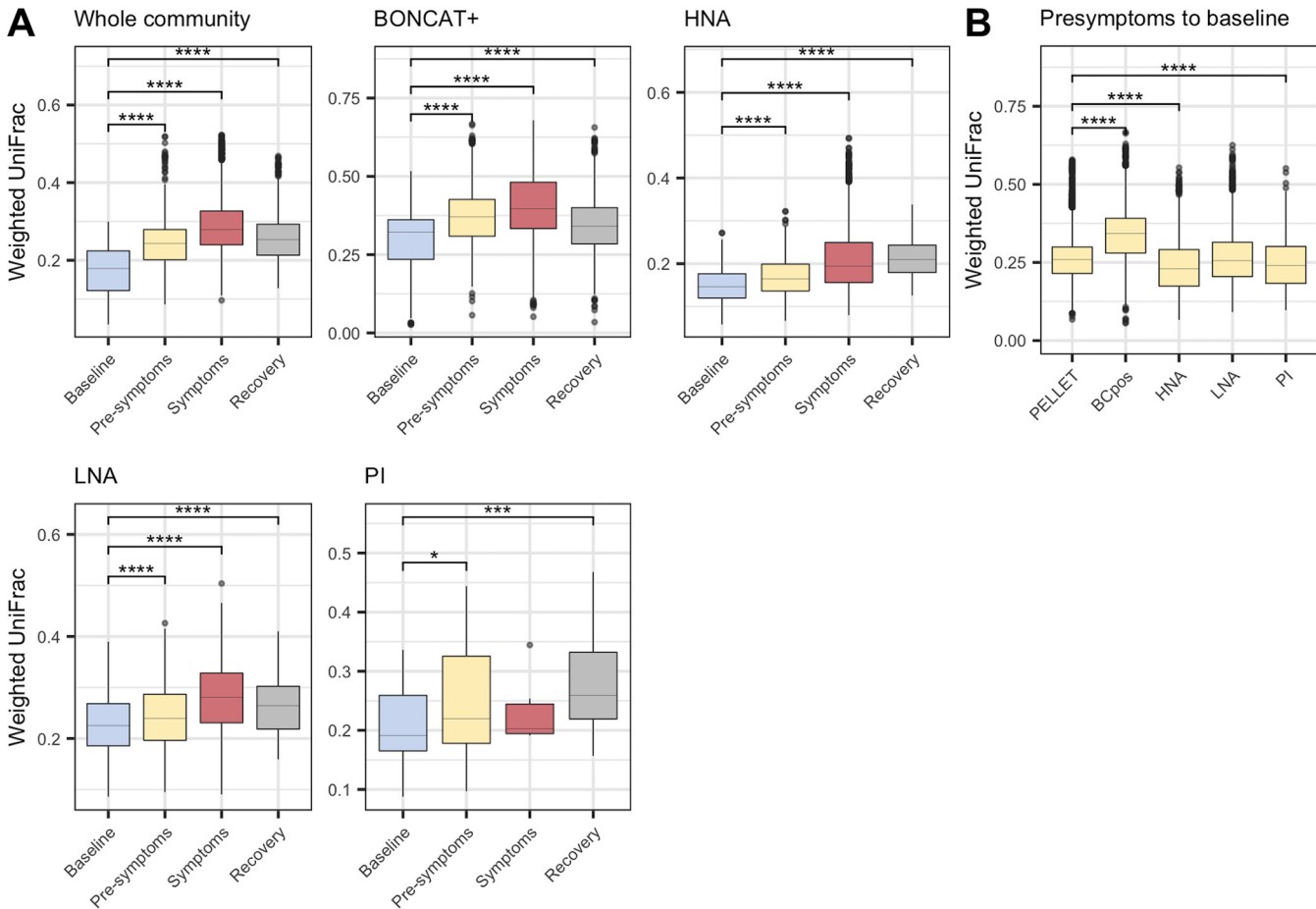

**FIG 4** Physiological fractions diverge in beta diversity from baseline. (A) Weighted UniFrac distances within each disease state per physiological fraction. Dunn's test for multiple comparisons to baseline. (B) Weighted UniFrac distances within each physiological fraction during the presymptomatic period, with Dunn's test for multiple comparisons to results for the whole community. *, $P < 0.05$, by PERMANOVA. Whole community, unsorted original sample; BONCAT, click-labeled protein-producing bacteria; HNA, high-nucleic-acid-content bacteria; LNA, low-nucleic-acid-content bacteria; PI, propidium iodide-stained bacteria with membrane damage.

abundant across physiological groups and disease states (using the most stringent cutoff of 0.9) (Table S2). These phylum- and genus-level dynamics are consistent with what has previously been described for DSS-induced colitis (16, 17).

As the BONCAT fraction changed the most in the presymptomatic disease state, we next set out to identify which bacteria changed their activity before disease onset (Fig. 6A). *Akkermansia*, the most prevalent genus within the *Verrucomicrobia* phylum, increased significantly in the BONCAT fraction as well as the whole-community, HNA, and LNA fractions (Fig. 6B; Table 2). Alongside the increase in *Akkermansia* organisms, *Bifidobacterium* (Fig. 6C), *Clostridiales vadin*, and an uncultured *Lachnospiraceae* organism (Fig. 6D and E) increased as well, while two members of the *Firmicutes* decreased in relative abundance (Fig. 6A).

Given the large changes in community composition at the phylum level, there were significant changes in the other physiological fractions occurring as well between the presymptomatic and baseline states. In the HNA fraction, 7 of 9 taxa that significantly decreased in abundance were members of the *Clostridiales*, and the only taxa that significantly increased were members of the *Clostridiales* and *Akkermansia*. In the LNA fraction, *Eubacterium* decreased and a *Ruminococcus* species decreased in the PI fraction. All differentially abundant taxa are depicted in Fig. S3. In the whole community, 12 taxa increased, including *Erysipelotrichaceae*, *Akkermansia*, and many *Firmicutes*. Twelve taxa decreased, five of which were members of the *Lachnospiraceae*. Overall, in multiple physiological states, taxa belonging to the *Clostridia* had variable responses,

**TABLE 1** PERMANOVA results for weighted UniFrac distances of the effect of physiology during each disease state

| Disease state | $R^2$ | P value |
|---|---|---|
| Baseline | 0.27 | 0.001 |
| Presymptomatic | 0.17 | 0.001 |
| Symptomatic | 0.14 | 0.001 |
| Recovery | 0.11 | 0.001 |
| Follow-up | 0.12 | 0.001 |

with some taxa increasing and others decreasing, emphasizing the variable response within the *Firmicutes* phylum, previously seen in DSS colitis (17).

## DISCUSSION

In this study, we applied a well-characterized perturbation resulting in dysbiosis to study the succession in bacterial physiology, as determined by nucleic acid content, membrane damage, and protein production of individual bacteria while still in their natural assemblage. Combining single-cell techniques and population-based sorting and sequencing (FACS-Seq), we report reproducible changes in these fractions prior to symptom onset in mice and before the gut microbiota reaches a dysbiotic stable state.

These changes are led by the translationally active bacteria, confirming that the gut microbiota is not homogeneous in its functional response to perturbations (16, 27, 29, 48, 49). Proportions of bacteria with membrane damage increase early on after DSS administration, indicating that the inflammatory intestinal environment is likely responsible, as this was not a direct cause of the microbial cells being exposed to DSS (see Fig. S4 in the supplemental material). The composition of this PI fraction did not change over time, suggesting that the same bacteria remain susceptible to damage. As described elsewhere (50–52), these bacteria are probably capable of membrane

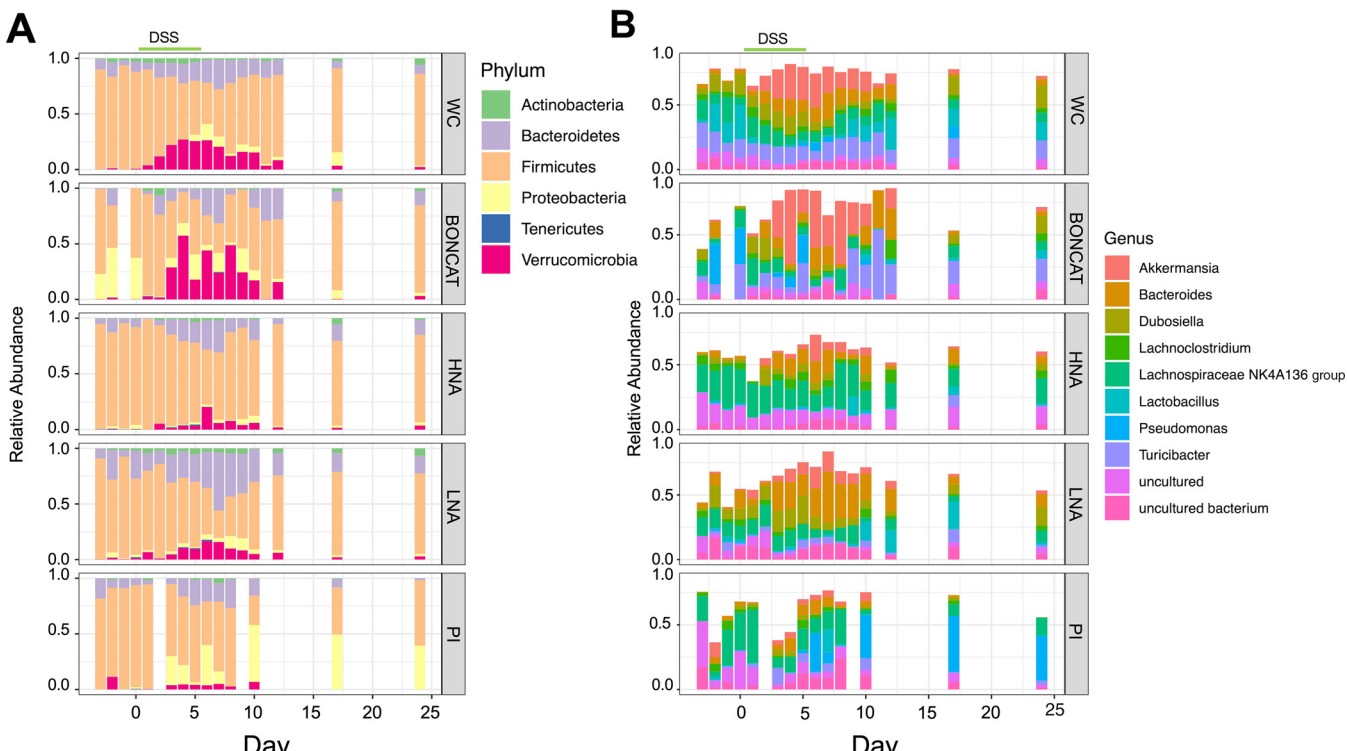

**FIG 5** Relative bacterial community composition in each physiological group over time. (A) Phylum level. (B) Genus level, with the top 10 genera plotted (*n* = 15) (3 experiments, 5 mice each). The period of DSS administration is highlighted with a green line. WC, whole community, the unsorted original sample; BONCAT, click-labeled translationally active bacteria; HNA, high-nucleic-acid-content bacteria; LNA, low-nucleic-acid-content bacteria; PI, propidium iodide-stained bacteria with membrane damage.

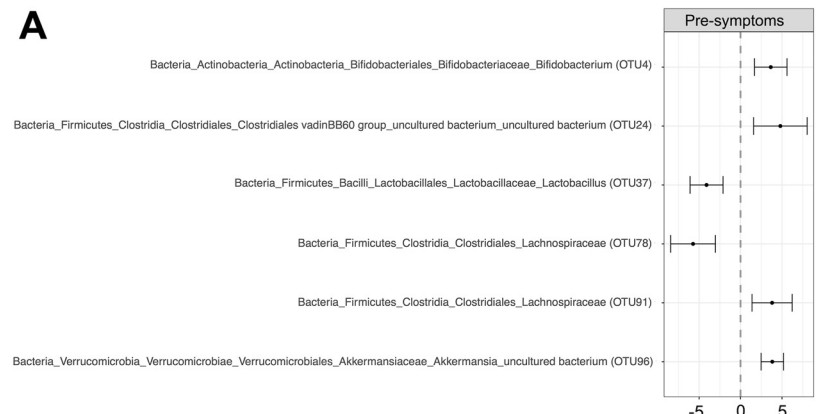

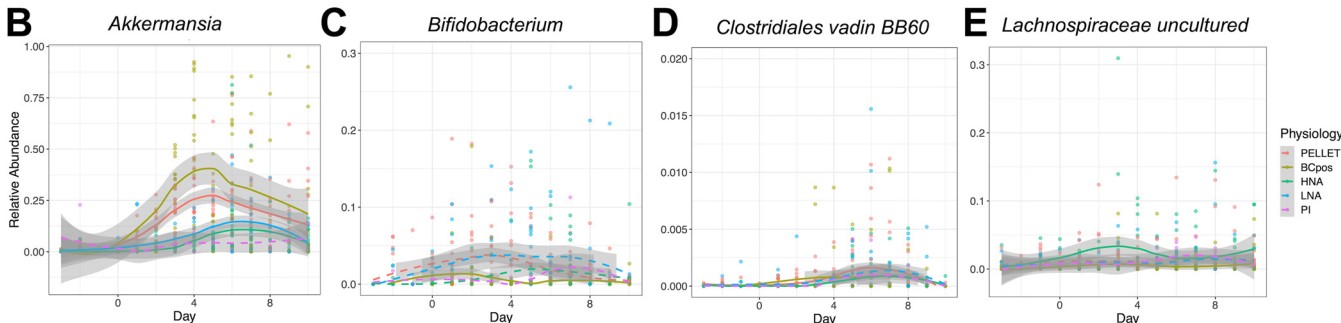

FIG 6 Differentially abundant taxa in the BONCAT+ fraction. (A) Differentially abundant taxa in the BONCAT+ fraction, comparing the presymptomatic disease state to baseline. Wald test, the significance cutoff is a *P* of <0.05 after false-discovery rate (FDR) correction. (B to D) Relative abundances over time of *Akkermansia* (B), *Bifidobacterium* (C), *Clostridiales Family XIII* (D), and *Lachnospiraceae* (E) organisms. Trends over time are shown with the local polynomial regression (loess) for each physiological fraction. Physiological fractions where the taxon was significantly differentially abundant are depicted as solid lines and where the taxon was not significantly differentially abundant between the presymptomatic period and baseline as dashed lines.

repair, ensuring that they remain in the gut ecosystem, as seen here. The decrease in the abundance of HNA bacteria is consistent with results for other systems in which the HNA bacteria are more susceptible to damage (27, 53–55), while the LNA bacteria have been found to be more resilient (55, 56). These differences in susceptibility to damage may be a taxonomic characteristic for gut microbial communities, as the *Firmicutes* have previously been suggested to be more susceptible to perturbations than the *Bacteroides* (57).

The increase in the proportion of *Akkermansia* seen in the translationally active fraction has previously been reported in DSS mouse models of colitis (15–17, 58). However, this increase is in direct contrast to data from patients with inflammatory bowel diseases (IBD), in whom *Akkermansia* is commonly depleted (59, 60). DSS is commonly used in mouse models of IBD, a chronic disease characterized by periods of acute inflammation interspersed with periods of remission. While the reasons for the

**TABLE 2** Relative abundance of *Akkermansia* in each physiological fraction in each disease state

| Physiology | % abundance during indicated disease state | | | |
| --- | --- | --- | --- | --- |
| | Baseline | Presymptomatic | Symptomatic | Recovery |
| Whole community | 0.37 | 14.16 | 23.37 | 13.57 |
| BONCAT+ | 0.97 | 24.94 | 33.89 | 16.04 |
| HNA | 0.67 | 2.85 | 8.48 | 5.98 |
| LNA | 0.70 | 5.00 | 12.62 | 5.06 |
| PI+ | 0.34 | 3.02 | 11.83 | 4.32 |
| Avg | 0.61 ± 0.26 | 9.99 ± 9.56 | 18.04 ± 10.48 | 8.99 ± 5.41 |

discrepancy in *Akkermansia* dynamics are unclear, differences between the DSS model of colitis and IBD are most likely at play. Functional differences between strains of *Akkermansia* in DSS colitis have been described (61) and may have an effect on disease outcome (14). Another possible explanation for the increased prevalence and activity of *Akkermansia* during DSS administration and colitis may be its competitive advantage during periods of low nutrient acquisition. The mice lose significant weight during colitis, with a decrease in nutrient acquisition and chow consumption. Previous studies on starvation report blooms of *Akkermansia*, possibly due to its ability to use mucin as a sole carbon and nitrogen source, providing it with a competitive advantage during low nutrient availability (62–65). There is increased oxygenation of the epithelium in DSS colitis (66), and *Akkermansia* has recently been shown to be able to adapt to these low levels of oxygen. These adaptations include an increased growth rate upon oxygen exposure, another possible competitive advantage against the luminal bacteria during inflammation (67). Lastly, a report on protein expression in a mouse model of colitis also demonstrated that *Akkermansia* had significantly increased protein expression during colitis compared to that at baseline. However, the increase in the relative abundance of *Akkermansia* was more mild (to 5%) than what we observed (an increase of 16.37%) (68). The role of *Akkermansia* in DSS-induced colitis is intriguing and warrants further study, as other reports have found that excessive mucin degradation may exacerbate colitis by allowing increased access to the epithelium and host immune system for other bacteria (69, 70).

Other bacteria that increased in the presymptomatic period include members of the *Ruminococcaceae* family. These include *Ruminiclostridium 5*, *Eubacterium*, and other undefined members. While the *Ruminococcaceae* family is typically decreased in IBD, members of the *Ruminococcaceae* family are mucin degraders, with *Ruminococcus gnavus* previously associated with IBD and known to produce a proinflammatory cytokine (71–74). The increased abundance and activity of *Bifidobacterium* have previously been reported in colitis as well (58, 75), yet this is surprising, as they are more commonly associated with health for their SCFA-producing properties (76, 77). The increase in the SCFA producers *Clostridiales Family XIII* is similarly interesting.

We have provided a detailed time series of gut microbial physiology during the progression to dysbiosis and recovery in DSS colitis. Previous longitudinal studies on DSS-induced colitis have focused either on active disease with minimal sampling before disease onset or on the recovery period by determining the effect of previous inflammatory episodes (16, 78, 79). A metatranscriptomics study that included one time point before symptom onset found minor changes in the transcriptional response of the gut microbiome (16). None of the changes before disease onset were significant, yet the trends were starting to appear, with downregulation of flagellar machinery and butyrate production in the *Clostridiales* and an increase in mucin-degrading enzymes in the *Bacteroidales* (16). Our work cannot discriminate which specific metabolic pathways are modified, but the more regular sampling allowed us to identify which bacteria are significantly altered, and how, before symptom onset. Importantly, the BONCAT technique can be expanded upon and combined with proteomics to glean insights into specific functionality. Limitations with PI as a marker of bacteria with membrane damage have been well discussed in previous work, as PI has been shown to also stain metabolically active cells. However, this appears to be limited, with no clear phylogenetic bias (36, 50, 52, 80, 81). Similarly, the biological complexities associated with nucleic acid content as a proxy for bacterial activity have been discussed elsewhere (49).

In the case of IBD, we hypothesize that the gut microbiota changes its activity and physiology in response to an environmental trigger, which then results in changes in composition detectable only at disease onset. As dysbiosis correlates with disease severity, we expect that changes in bacterial physiology and activity would occur before the onset of a flare and may represent a possible window of intervention or an early diagnostic tool. Here, we demonstrate how these are reproducible changes, which can be monitored in a rapid and efficient manner by using flow cytometry and monitoring

relative nucleic acid content, protein production, and membrane damage. Critically, many of these physiological changes would have been missed by traditional sequencing approaches. Monitoring of the translationally active bacteria identified taxa that increased or decreased their activity levels and, thus, are potential therapeutic targets. Indeed, through the targeted improvement of bacterial metabolism, the progression to dysbiosis may be minimized. Specifically, the active role of *Akkermansia* in disease progression, which seems to thrive in the inflammatory milieu induced by DSS, warrants further investigation.

**Conclusions.** We report a clear and reproducible succession in bacterial physiology in response to a DSS-induced perturbation; an increase in bacteria with membrane damage occurs before the onset of symptoms, concomitantly with strong changes in the diversity of the translationally active bacteria. These bacteria may become potential biomarkers of an upcoming dysbiosis, allowing for mitigation interventions before dysbiosis sets in and the gut microbiota reaches a new alternative stable state. Specifically in DSS-induced colitis, many of these changes in community composition and diversity are driven largely by *Akkermansia*. Overall, this work demonstrates the use of single-cell and population-based methods to identify changes in the gut microbiome otherwise missed by whole-community sequencing and cross-sectional surveys. By identifying functional changes in bacterial physiology prior to disease onset, this work furthers the goals of targeted gut microbiome therapies.

## MATERIALS AND METHODS

**DSS-induced colitis mouse model.** Wild-type C57/BL6 mice were purchased from Jackson Laboratory at 5 weeks of age and left to acclimatize to our animal facility for 2 weeks before the start of the experiment. Colitis was induced with 2% dextran sodium sulfate (DSS) in drinking water for 5 days (molecular weight, 36 to 50 kDa; MP Biomedicals). Mice were housed under specific-pathogen-free conditions at the Goodman Cancer Center at McGill University (animal ethics protocol 2018-7999). The experimental setup was independently completed 3 times, with 5 mice per cage each time.

The disease activity index was assessed based on the presence of blood in stool, stool consistency, and change in body weight (82). The presence of blood in stool was assessed daily for each mouse using the Hemoccult Sensa kit (Beckman Coulter). Body weight was measured daily. Fecal lipocalin 2 was assessed with the mouse lipocalin 2 DuoSet enzyme-linked immunosorbent assay (ELISA) (R&D). Samples were prepared according to the method of Chassaing et al. (83) and diluted 2-fold to 2,000-fold depending on experimental day (83). ELISA was performed according to the manufacturer's instructions.

**Fecal sample preparation and BONCAT.** Fecal samples were collected from mice daily and transferred to anaerobic conditions within 1 h of collection. Sample preparation was performed in an anaerobic chamber (Coy Laboratory Products; 5% $H_2$, 20% $CO_2$, 75% $N_2$); flow cytometry acquisition and cell sorting were performed aerobically. Gut microbiota sample preparation was carried out as previously described (80). For the bioorthogonal noncanonical amino acid tagging (BONCAT) incubations, bacteria were diluted 1/10 in 50% (vol/vol) of the supernatant retained from the first 6,000 $\times$ $g$ centrifugation and the remaining volume of reduced phosphate-buffered saline (PBS). Bacteria were incubated at 37°C for 2 h with a 2 mM final concentration of L-homopropargylglycine (HPG). A no-HPG incubation control was included, and each sample was incubated in duplicate. Bacteria were fixed with 80% ethanol to a final concentration of 50% (vol/vol) and stored at 4°C until processed with the click reaction that same day.

For the click reaction, bacteria were pelleted and resuspended in the click reaction solution (Click-iT Cell buffer kit; ThermoFisher Scientific) containing 5 $\mu$M Alexa-647 azide and incubated in the dark at room temperature for 30 min. A no-Alexa-647 azide control was included. Samples were then centrifuged at 8,000 $\times$ $g$ for 5 min, washed with 80% ethanol, and resuspended in PBS for flow cytometry acquisition and cell sorting.

**Flow cytometry acquisition and cell sorting.** Acquisition to determine cell concentrations and proportions of different bacterial physiologic fractions was performed on a BD FACSCanto II equipped with a 488-nm laser and 530/30 and 585/42 detection filters. Fecal samples were collected from mice daily and transferred to anaerobic conditions within 1 h of collection. For each mouse, the freshly collected fecal samples are put in solution and split into different FACS tubes for individual staining to minimize stain interference and overlap. Cells are either stained with SYBR green I (Invitrogen; 1$\times$ final concentration) for 15 min to detect the more active (high nucleic acid content [HNA]) and less active (low nucleic acid content [LNA]) physiological fractions or with propidium iodide (Sigma) for 10 min (0.08 mg · ml$^{-1}$ final concentration) to detect membrane damage. All staining was performed in the dark under anaerobic conditions.

For flow cytometry acquisition, rainbow fluorescent particles of 3.0 to 3.4 $\mu$m (BD Biosciences) were added to each sample before acquisition in sufficient volume (10 to 30 $\mu$l) to acquire bead events equivalent to ~1% of total events. Rainbow fluorescent particle concentration and total counts were determined after acquisition with 7-$\mu$m CountBright absolute counting beads (Life Technologies), as per previous studies (80). Cell sorting was performed on a FACSAria III (BD Bioscience) equipped with a 488-nm laser and the appropriate detection filters, using a 70-$\mu$m nozzle at 70 lb/in$^2$ and at a flow rate that would lead to less than 5% coincidence events. Positively stained cells were determined from debris and unstained cells using unstained controls. One hundred eighty thousand events were sorted using a 70-$\mu$m nozzle for each population in each individual and frozen at −80°C for later DNA extraction. Sheath

fluid was collected at the end of every sorting day as a negative control to detect contaminant DNA. Data files were analyzed using FlowJo V7 software (FlowJo LLC). Cell count data were analyzed as previously reported (80). Cell count and proportional data were analyzed with a mixed-effects analysis model with repeated measures.

**DNA extraction and 16S rRNA gene amplicon bioinformatics analysis.** The V4-V5 hypervariable region was amplified with the 515F/926R primers (84). Preprocessing was performed with the quantitative insights into microbial ecology (QIIME2) platform (85). Trimming, alignment of paired-end reads, and quality filtering were performed by DADA2 per sequencing run (86). Taxonomic alignment was performed with a pretrained naive Bayes classifier using the SILVA 132 database on 99% operational taxonomic units (OTUs). Prevalence-based filtering was done per experiment. Reads present in fewer than 3 samples and reads present fewer than 10 times were removed. Reads present in the sheath fluid but absent in the whole-community samples were identified as contaminants and removed. Sequencing runs were then merged.

Bioinformatics analysis was performed with phyloseq (v1.3) in R (v3.6.1). Relative abundance data were used for beta diversity analysis. Beta diversity was assessed on weighted UniFrac distances calculated using rbiom (v1.0). Pairwise PERMANOVAs were calculated with 999 permutations to test for significance using adonis in the vegan package (v2.5). Differential-abundance testing was performed after phylogenetically agglomerating taxa based on a phylogenetic tree length of 0.1. The statistical analysis package corncob (v0.1) was used for differential abundance testing, which performs beta-binomial regression models to determine differentially abundant and dispersed relative abundances (87). Beta diversity dissimilarities between physiological groups or disease states were compared using the Kruskal-Wallis test with Benjamini-Hochberg (BH) correction for multiple comparisons in the rstatix package (v0.5). Differential-abundance testing across all groups was performed with ANCOM II, with BH correction for multiple comparisons (88).

**Ethics.** The study was approved by the McGill Ethics Research Board (animal protocol 2018-7999), Montreal, QC, Canada.

**Data availability.** Bacterial 16S rRNA gene sequencing data can be accessed in the SRA database under accession number PRJNA719860. Code related to the analysis has been deposited in GitHub (https://github.com/MTaguer).

## SUPPLEMENTAL MATERIAL

Supplemental material is available online only.
**FIG S1,** PDF file, 0.1 MB.
**FIG S2,** PDF file, 0.2 MB.
**FIG S3,** PDF file, 0.1 MB.
**FIG S4,** PDF file, 0.6 MB.
**TABLE S1,** Excel file, 0.02 MB.
**TABLE S2,** Excel file, 0.01 MB.

## ACKNOWLEDGMENTS

We thank members of the Maurice lab for their aid in editing parts of the manuscript. As well, we thank Josep Gasol, Marta Sebastian, and E. C. Marga for their useful advice on the BONCAT procedure.

We declare that we have no competing interests.

This work was supported by a Natural Sciences and Engineering Research Council (NSERC) fellowship awarded to M.T. (PGSD3-504903-2017) and a discovery grant from NSERC for A.T.-C. This work was also supported by the Canada Research Chair program (grant 950-230748 X-242502), a Canadian Institutes of Health Research (CIHR) transition grant to C.F.M. (PJT-149098), and the Azrieli Global Scholar Program from the Canadian Institute for Advanced Research (CIFAR) to C.F.M. The flow cytometry work was performed in the Flow Cytometry Core Facility for flow cytometry, and single-cell analysis was performed in the Life Science Complex and supported by funding from the Canadian Foundation for Innovation.

M.T. designed, performed, analyzed, and interpreted experiments. E.D. and K.W. aided in flow cytometry experiments. A.T.-C. and D.A.S. performed the Bayesian analysis. C.F.M. conceived the project, obtained funding, helped design experiments, and helped to interpret the data. All authors edited the manuscript and approved the final draft.

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
