## [Reviewer comments · mSystems]

Changes in gut bacterial translation occur before symptom onset and dysbiosis in dextran sodium sulfate-induced murine colitis

Mariia Taguer, Emma Darbinian, Kailee Wark, Alicia Ter-Cheam, David Stephens, and Corinne Maurice

Corresponding Author(s): Corinne Maurice, McGill University

Review Timeline:

Submission Date:	April 23, 2021
Editorial Decision:	July 14, 2021
Revision Received:	August 23, 2021
Editorial Decision:	September 11, 2021
Revision Received:	October 11, 2021
Accepted:	October 20, 2021

Editor: Mariana Byndloss

Reviewer(s): The reviewers have opted to remain anonymous.

Transaction Report:

DOI: <https://doi.org/10.1128/mSystems.00507-21>

July 14, 2021

Dr. Corinne F Maurice
McGill University
Microbiology & Immunology
Montreal, QC H3G 0B1
Canada

Re: mSystems00507-21 (Changes in gut bacterial translation occur before symptom onset and dysbiosis in dextran sodium sulfate-induced murine colitis)

Dear Dr. Corinne F Maurice:

Thank you for submitting your manuscript to mSystems. We have completed our review and I am pleased to inform you that, in principle, we expect to accept it for publication in mSystems. However, acceptance will not be final until you have adequately addressed the reviewer comments.

Preparing Revision Guidelines

For complete guidelines on revision requirements for your article type, please see the journal Article Types requirement at <https://journals.asm.org/journal/mSystems/article-types>. **Submissions of a paper that does not conform to mSystems guidelines will delay acceptance of your manuscript.**

Sincerely,

Mariana Byndloss

Editor, mSystems

Journals Department
Reviewer comments:

Reviewer #1 (Comments for the Author):

Summary:

Taguer et al. sought to assess physiologic gut microbiota changes over time in response to intestinal disease/perturbation using a DSS model of colitis in mice. Specifically, the authors determined how the diversity of three physiologically distinct aspects of the gut microbiota following perturbation induced by DSS colitis by measuring bacterial relative nucleic acid content, membrane damage, and protein production. Given how dynamic the gut microbiome is when perturbed, the authors' focus of following gut microbial community transitions longitudinally (before, during, and after perturbation) induced by intestinal disease is a major strength of this study. Longitudinal studies such as this are important endeavors in order to continue to make progress and move the microbiome field forward.

While the research premise and experimental design are strong, there is room for clarification in several areas of the manuscript. The FACS methodology is not clear regarding whether the 3 functional aspects of gut bacteria (nucleic acid content, membrane damage, and protein production) were stained and sorted utilizing the same fecal sample and, furthermore, whether all the staining for these functional aspects were performed in a single test tube or multiple tubes for each functional category. The use of the term "fraction" implies that all staining and sorting to explore i) nucleic acid content, ii) membrane damage, and iii) protein production was all done in a single tube. Clarification would be essential here.

Major comments:

1. It is initially confusing to know what the authors mean when saying "functional succession of bacterial communities". Please elaborate or be more specific. Would "sequential functional changes of bacterial communities" or "longitudinal functional changes of bacterial communities", or a similar re-phrasing, be more accurate?
2. While the authors briefly touch on limitations of the work done here, it is suggested that additional care and detail is included to discuss and consider the limitations of these techniques and their interpretation of the results to evaluate gut bacterial function when appropriate. For example, is assessing the membrane damage identified through membrane exclusion dyes appropriate for gram positives bacteria that lack an outer membrane? Furthermore, how were dead bacterial cells excluded in flow cytometry analyses? How are live bacterial cells with membrane damage distinguished from intact dead bacterial cells here? Dead cells that are still intact would not be excluded by gating out debris, so this would not be sufficient.
3. Do the data support that microbial functional alterations allow for an expansion of *Akkermansia muciniphila*? This may be too strong of a conclusion.
4. The terminology of "single-cell techniques" used throughout the manuscript should be addressed. To say that single-cell techniques were used in this manuscript is misleading. The cell sorting done using FACS was population based, not single-cell based. Populations of cells were subsequently analyzed, not analysis of single cells.

Minor Comments:

1. Key words as abbreviations should be avoided when possible (specifically HNA and LNA).
2. Line 38: The vagueness of the phrase "damaged bacteria and diversity changes in the protein-producing fraction" is confusing. When first encountered in the abstract, it is unclear what is meant by referring to a fraction of microbiota as "protein-producing bacteria" since all bacteria produce proteins. Perhaps using the phrase "translationally active bacteria", as the authors use in parts of the manuscript, would be preferable in the abstract and throughout. Similarly, "damaged bacteria" could be described more specifically (perhaps simply by referring to the bacterial membrane damage).
3. Lines 62-63: It is encouraged that more recent, key publications supporting this statement also be cited here.
4. Lines 74-75: Please provide citations to support the statement that microbiomes not in steady state are most amenable to modulation.
5. It is suggested that Fig 1B and 1E can be moved to supplemental.
6. Fig 2C: Should the black square above the symptoms state be a star to indicate significance/p-value? Or does this symbol mean something else?
7. Fig 3: Define W1 and W2 in legend or expand on figure.
8. Do the authors suppose that the bacteria with damaged membranes recover, or do they likely die? Could the authors discuss the consequence of the membrane damage, to the bacterial cell and to the microbiome or host outcomes?

Reviewer #3 (Comments for the Author):

This study describes the use of a suite of interesting and useful methods to assess the physiology of bacteria involved in the succession of microbes that occur in a longitudinal study of DSS-induced colitis. This project has considerable potential. Unfortunately, the underlying data (Figure 1) and the way the underlying data were analyzed has some problems which are described below.

MAJOR COMMENTS

1. Concerning the comment above about Figure 1, the authors wrote the following.

"The onset of colitis was consistent across experiments based on weight loss (Fig. 1c), increases in fecal lipocalin-2 levels (Fig. 1d), and the presence of blood in stool (Fig. 1e), with some cage-level variation."

When this reviewer examines those figures, the onset of colitis does not seem consistent as assessed by weight and the presence of blood in the feces.

For weight for Cage A, there is a gradual decrease in weight starting on Day 4 and lasting until Day 9 and then the weights increased. For Cage B, there is a gradual decrease in weight starting on Day 5 until Day 8 and then the weights increased. For Cage C, there is a large decrease in weights from Day 3 to Day 4 and then the weights increased.

For blood in the feces for Cage A, there are four scattered and unlinked (to the subsequent day) detections of blood in Days 1-4 and then four of the five mice start to have consistent blood on Day 6 but mouse M9 never had blood in its feces. For Cage B, for two of the mice, consistent blood in the feces starts on Day 3 through ~Day 7 while the other three mice started on Day 4 through ~Day 7. For Cage C, consistent blood in the feces starts on Day 3 for one mouse, Day 4 for three mice, and Day 5 for one mouse.

Thus, this reviewer does not understand how the authors could make that statement about the consistency of these results.

2. Making matters worse, based on these data, the authors then grouped the samples by day for all of the subsequent analyses. However, since there is considerable variation by day for the DSS symptoms as described above, this is going to lead to results that are muddled by all of this variation, which is what this reviewer saw in much of the subsequent data analyses, as demonstrated by the large error bars.

3. This reviewer thinks that the downstream data analyses would have produced much clearer results if the authors grouped the animals by the expression of symptoms - e.g. weight and blood in feces - instead of by day.

4. The author's also did not perform a vehicle control, which is important, even for longitudinal studies.

5. The author's also did not repeat the experiment. It appears that this experiment was performed only once.

MINOR COMMENTS

1. The term diversity has some very specific meanings and it is used inappropriately in numerous places in the manuscript. For example, in the sentence below (Lines 84-86), since diversity does not mean specific taxa, this is not an appropriate use - "composition" would be a better in this particular sentence.

"The changes in bacterial diversity are consistent, with decreases in the short-chain fatty acid (SCFA)-producers Clostridiales, and increases in the more oxygen-tolerant and pro-inflammatory Enterobacteriaceae[11,12]."

"Changes in gut bacterial translation occur before symptom onset and dysbiosis in dextran sodium sulfate-induced murine colitis"

Manuscript #: mSystems00507-21

Summary:

Taguer *et al.* sought to assess physiologic gut microbiota changes over time in response to intestinal disease/perturbation using a DSS model of colitis in mice. Specifically, the authors determined how the diversity of three physiologically distinct aspects of the gut microbiota following perturbation induced by DSS colitis by measuring bacterial relative nucleic acid content, membrane damage, and protein production. Given how dynamic the gut microbiome is when perturbed, the authors' focus of following gut microbial community transitions longitudinally (before, during, and after perturbation) induced by intestinal disease is a major strength of this study. Longitudinal studies such as this are important endeavors in order to continue to make progress and move the microbiome field forward.

While the research premise and experimental design are strong, there is room for clarification in several areas of the manuscript. The FACS methodology is not clear regarding whether the 3 functional aspects of gut bacteria (nucleic acid content, membrane damage, and protein production) were stained and sorted utilizing the same fecal sample and, furthermore, whether all the staining for these functional aspects were performed in a single test tube or multiple tubes for each functional category. The use of the term "fraction" implies that all staining and sorting to explore i) nucleic acid content, ii) membrane damage, and iii) protein production was all done in a single tube. Clarification would be essential here.

Major comments:

1. It is initially confusing to know what the authors mean when saying "functional succession of bacterial communities". Please elaborate or be more specific. Would "sequential functional changes of bacterial communities" or "longitudinal functional changes of bacterial communities", or a similar re-phrasing, be more accurate?
2. While the authors briefly touch on limitations of the work done here, it is suggested that additional care and detail is included to discuss and consider the limitations of these techniques and their interpretation of the results to evaluate gut bacterial function when appropriate. For example, is assessing the membrane damage identified through membrane exclusion dyes appropriate for gram positives bacteria that lack an outer membrane? Furthermore, how were dead bacterial cells excluded in flow cytometry analyses? How are live bacterial cells with membrane damage distinguished from intact dead bacterial cells here? Dead cells that are still intact would not be excluded by gating out debris, so this would not be sufficient.
3. Do the data support that microbial functional alterations allow for an expansion of *Akkermansia muciniphila*? This may be too strong of a conclusion.
4. The terminology of "single-cell techniques" used throughout the manuscript should be addressed. To say that single-cell techniques were used in this manuscript is misleading. The cell sorting done using FACS was population based, not single-cell based. Populations of cells were subsequently analyzed, not analysis of single cells.

Minor Comments:

1. Key words as abbreviations should be avoided when possible (specifically HNA and LNA).
2. Line 38: The vagueness of the phrase "damaged bacteria and diversity changes in the protein-producing fraction" is confusing. When first encountered in the abstract, it is unclear what is meant by referring to a fraction of microbiota as "protein-producing bacteria" since all bacteria produce proteins. Perhaps using the phrase "translationally active bacteria", as the authors use in parts of the manuscript, would be preferable in the abstract and throughout. Similarly, "damaged bacteria" could be described more specifically (perhaps simply by referring to the bacterial membrane damage).

3. Lines 62-63: It is encouraged that more recent, key publications supporting this statement also be cited here.
4. Lines 74-75: Please provide citations to support the statement that microbiomes not in steady state are most amendable to modulation.
5. It is suggested that Fig 1B and 1E can be moved to supplemental.
6. Fig 2C: Should the black square above the symptoms state be a star to indicate significance/p-value? Or does this symbol mean something else?
7. Fig 3: Define W1 and W2 in legend or expand on figure.
8. Do the authors suppose that the bacteria with damaged membranes recover, or do they likely die? Could the authors discuss the consequence of the membrane damage, to the bacterial cell and to the microbiome or host outcomes?

Response to Reviewers

"Changes in gut bacterial translation occur before symptom onset and dysbiosis in dextran sodium sulfate-induced murine colitis"

Manuscript #: mSystems00507-21

Reviewer #1 (Comments for the Author):

Summary:

Taguer et al. sought to assess physiologic gut microbiota changes over time in response to intestinal disease/perturbation using a DSS model of colitis in mice. Specifically, the authors determined how the diversity of three physiologically distinct aspects of the gut microbiota following perturbation induced by DSS colitis by measuring bacterial relative nucleic acid content, membrane damage, and protein production. Given how dynamic the gut microbiome is when perturbed, the authors' focus of following gut microbial community transitions longitudinally (before, during, and after perturbation) induced by intestinal disease is a major strength of this study. Longitudinal studies such as this are important endeavors in order to continue to make progress and move the microbiome field forward.

We thank the reviewer for highlighting the importance of longitudinal studies for microbiome work.

While the research premise and experimental design are strong, there is room for clarification in several areas of the manuscript. The FACS methodology is not clear regarding whether the 3 functional aspects of gut bacteria (nucleic acid content, membrane damage, and protein production) were stained and sorted utilizing the same fecal sample and, furthermore, whether all the staining for these functional aspects were performed in a single test tube or multiple tubes for each functional category. The use of the term "fraction" implies that all staining and sorting to explore i) nucleic acid content, ii) membrane damage, and iii) protein production was all done in a single tube. Clarification would be essential here.

We apologize for the lack of clarity. For each mouse and on each day, the same fecal pellet collected was used for all stains, but staining was performed in separate FACS tubes to minimize stain interference and overlap (Stiefel et al., BMC Microbiology, 2015,

15; Shi et al., Cytometry A, 2007, 71 (8)). Thus, sorted cells come from individually stained tubes from the same original fecal sample. This is now explicitly stated on L180-186: *“Fecal samples were collected from mice daily and transferred into anaerobic conditions within one hour of collection. For each mouse, the freshly collected fecal samples are put in solution and split into different FACS tubes for individual staining to minimize stain interference and overlap. Cells are either stained with SYBR Green I (Invitrogen, 1X final concentration) for 15 minutes to detect the more active (high nucleic acid content - HNA) and less active (low nucleic acid content – LNA) physiological fractions, or with propidium iodide (Sigma) for 10 min (0.08mg.ml⁻¹ final concentration) to detect membrane damage.”*

Major comments:

1. It is initially confusing to know what the authors mean when saying "functional succession of bacterial communities". Please elaborate or be more specific. Would "sequential functional changes of bacterial communities" or "longitudinal functional changes of bacterial communities", or a similar re-phrasing, be more accurate?

We have updated this sentence following the reviewer's suggestion to *“Determining the longitudinal functional changes of bacterial communities”* (L111).

2. While the authors briefly touch on limitations of the work done here, it is suggested that additional care and detail is included to discuss and consider the limitations of these techniques and their interpretation of the results to evaluate gut bacterial function when appropriate. For example, is assessing the membrane damage identified through membrane exclusion dyes appropriate for gram positives bacteria that lack an outer membrane? Furthermore, how were dead bacterial cells excluded in flow cytometry analyses? How are live bacterial cells with membrane damage distinguished from intact dead bacterial cells here? Dead cells that are still intact would not be excluded by gating out debris, so this would not be sufficient.

We thank the reviewer for this important comment about flow cytometry analyses.

PI has been optimized and used with a variety of bacterial strains and microbial communities to assess membrane damage, and has been shown to be appropriate with both Gram + and Gram - cells (Shi et al., Cytometry A, 2007, 71 (8); Feng et al., Front Med, 2018, 5; Buysschaert et al., Cytometry A, 2018, 93 (2); Maurice et al., Methods Enzymol, 2013; Rosenberg et al., Sci Rep, 2019, 9). Several studies also show that PI can enter viable cells as soon as membrane integrity is altered, such as during cell replication, or even when cells are metabolically active (Shi et al., Cytometry A, 2007, 71 (8); Davey & Hexley, Environ Microbiol, 2011, 13 (1); Trevors, J Microbiol Meth, 2012, 90 (1); Rosenberg et al., Sci Rep, 2019, 9). These studies all emphasize the importance of confirming PI final concentrations according to the bacterial sample used,

and to include killed controls. We have proceeded with both, as stated in our methods (L195-196).

We are unsure what the reviewer means by “excluding dead bacterial cells from our analyses”, and assume the reviewer is referring to excluding PI+ cells. The individual staining approach we used does not allow for the removal of PI+ cells in our flow cytometry analyses, but our FACSeq approach allows us to identify them in our downstream analyses. We thus were able to identify cells with loss of membrane integrity but did not exclude them from any of our analyses, as one of our goals was to determine if the identity and proportions of damaged cells changed during the progression to colitis. Excluding these cells from our analyses would not have allowed us to explore this.

For all our flow cytometry analyses, bacterial cells of interest were identified using both side scatter (SSC, providing information on cell granularity and size) and fluorescence values according to the dye used (FL1 for SybrGreen 1, FL3 for PI and BONCAT). Density plots were used to gate bacterial cells to determine the proportions of cells within each physiological category. Cell debris are therefore automatically excluded, as they don't have detectable SSC or fluorescence values. As SybrGreen 1 and PI are both nucleic acid stains, dead cells lacking DNA would not be stained and therefore excluded from our flow cytometry analysis.

Nevertheless, we have added some details and provided additional references about the limitations of these physiological stains in the Discussion (L 400-404).

3. Do the data support that microbial functional alterations allow for an expansion of *Akkermansia muciniphila*? This may be too strong of a conclusion.

We agree with the reviewer that 16S data does not allow for absolute quantification and have modified the text to refer to increases in relative abundances throughout this revised version.

4. The terminology of "single-cell techniques" used throughout the manuscript should be addressed. To say that single-cell techniques were used in this manuscript is misleading. The cell sorting done using FACS was population based, not single-cell based. Populations of cells were subsequently analyzed, not analysis of single cells. We agree with the reviewer that our approach was not a single-cell sequencing approach and was population-based. Yet, flow cytometry is a single cell tool, allowing to get information on individual cells in a mixed community, which we have used for the quantification of physiological categories (HNA, LNA, PI+, and BONCAT). We have

therefore adjusted our wording throughout the manuscript to reflect that our cell sorting was population-based, and that our physiological analysis was single-cell based.

Minor Comments:

1. Key words as abbreviations should be avoided when possible (specifically HNA and LNA).

This has been corrected.

2. Line 38: The vagueness of the phrase "damaged bacteria and diversity changes in the protein-producing fraction" is confusing. When first encountered in the abstract, it is unclear what is meant by referring to a fraction of microbiota as "protein-producing bacteria" since all bacteria produce proteins. Perhaps using the phrase "translationally active bacteria", as the authors use in parts of the manuscript, would be preferable in the abstract and throughout. Similarly, "damaged bacteria" could be described more specifically (perhaps simply by referring to the bacterial membrane damage).

We apologize for the confusion and have used the terms "translationally active bacteria" and "bacteria with membrane damage" in this revised version of our manuscript.

3. Lines 62-63: It is encouraged that more recent, key publications supporting this statement also be cited here.

Three more recent papers have been added.

4. Lines 74-75: Please provide citations to support the statement that microbiomes not in steady state are most amenable to modulation.

Additional citations have been added.

5. It is suggested that Fig 1B and 1E can be moved to supplemental.

This has been done.

6. Fig 2C: Should the black square above the symptoms state be a star to indicate significance/p-value? Or does this symbol mean something else?

This has been updated to have the p-value written out for clarity.

7. Fig 3: Define W1 and W2 in legend or expand on figure.

This has been clarified in the legend.

8. Do the authors suppose that the bacteria with damaged membranes recover, or do they likely die? Could the authors discuss the consequence of the membrane damage, to the bacterial cell and to the microbiome or host outcomes?

This is a very interesting point, which we can only speculate about with our data. There is evidence that PI+ cells are metabolically active and can recover (Shi et al., Cytometry A, 2007, 71 (8); Rosenberg et al., Sci Rep, 2019, 9); yet we are not aware that this has been quantified for bacterial cells in complex communities. In yeast, it was estimated that 7% of damaged cells could repair after chemical stress (Davey & Hexley, Environ Microbiol, 2011, 13 (1)), and more recently, the bacterium *Shewanella decolorationis* was shown to be able to repair its membrane through metabolic modifications (Yang et al., Sci rep, 2015, 5).

Given the strong selective forces on gut microbial communities (high bacterial density, gut peristaltic movement, high levels of circulating immune cells and antimicrobial peptides, etc.), we speculate that most damaged bacterial cells die and are removed from the gut ecosystem. This would explain the strong community structure changes commonly observed in disease. Important bacterial death and high levels of cellular debris would be immunogenic, as this would result in high levels of circulating LPS and other pro-inflammatory molecules.

In contrast, if damaged bacterial cells were to mostly repair, then most metabolic resources would be allocated to bacterial repair and metabolite production (such as short-chain fatty acid production, for example) (Ferenci, Trends Microbiol, 2016, 24 (3)). This would be consistent with changes in bacterial metabolism without changes in bacterial community structure, as has been observed during periods of stress or remission of inflammatory conditions (Becattini et al., Cell Host Microbe, 2021, 29; Schirmer et al., Nat Microbiol, 2018, 3 (3))

However, as this is mostly speculative, we have not added anything in our Discussion to address this.

Reviewer #3 (Comments for the Author):

This study describes the use of a suite of interesting and useful methods to assess the physiology of bacteria involved in the succession of microbes that occur in a longitudinal study of DSS-induced colitis. This project has considerable potential. Unfortunately, the underlying data (Figure 1) and the way the underlying data were analyzed has some problems which are described below.

We thank the reviewer for the positive comments and hope to have addressed their major concerns about our data and data analysis below.

MAJOR COMMENTS

1. Concerning the comment above about Figure 1, the authors wrote the following.

"The onset of colitis was consistent across experiments based on weight loss (Fig. 1c), increases in fecal lipocalin-2 levels (Fig. 1d), and the presence of blood in stool (Fig. 1e), with some cage-level variation."

When this reviewer examines those figures, the onset of colitis does not seem consistent as assessed by weight and the presence of blood in the feces.

For weight for Cage A, there is a gradual decrease in weight starting on Day 4 and lasting until Day 9 and then the weights increased. For Cage B, there is a gradual decrease in weight starting on Day 5 until Day 8 and then the weights increased. For Cage C, there is a large decrease in weights from Day 3 to Day 4 and then the weights increased.

For blood in the feces for Cage A, there are four scattered and unlinked (to the subsequent day) detections of blood in Days 1-4 and then four of the five mice start to have consistent blood on Day 6 but mouse M9 never had blood in its feces. For Cage B, for two of the mice, consistent blood in the feces starts on Day 3 through ~Day 7 while the other three mice started on Day 4 through ~Day 7. For Cage C, consistent blood in the feces starts on Day 3 for one mouse, Day 4 for three mice, and Day 5 for one mouse.

Thus, this reviewer does not understand how the authors could make that statement about the consistency of these results.

We thank the reviewer for this excellent suggestion. We have now proceeded to determine the Disease Activity Index (DAI) of each mouse to determine the different colitis conditions. The DAI considers the presence of blood in stools, the degree of weight loss, and stool consistency (Kim et al., J Vis Exp, 2012, 60 (3678)), and is now found in Figure 1 and Supplemental Table 1. This index was produced per mouse and for each sample collection day. We complemented the DAI with lipocalin 2 assays which were present in our original submission, to better characterize the 4 disease states (baseline, pre-symptoms, symptoms, recovery). We have modified our Methods and Results sections accordingly.

2. Making matters worse, based on these data, the authors then grouped the samples by day for all of the subsequent analyses. However, since there is considerable variation by day for the DSS symptoms as described above, this is going to lead to results that are muddled by all of this variation, which is what this reviewer saw in much of the subsequent data analyses, as demonstrated by the large error bars.

We agree and have re-analyzed our data as detailed below and suggested by the reviewer.

3. This reviewer thinks that the downstream data analyses would have produced much clearer results if the authors grouped the animals by the expression of symptoms - e.g. weight and blood in feces - instead of by day.

We thank the reviewer for this suggestion and have updated our analysis to reflect the grouping of animals by disease activity index (DAI) and lipocalin 2 assays, rather than using the average day of symptoms onset.

This has slightly changed the timing of the 4 disease states for each mouse, but we see similar results as our original submission and analysis: the decrease in PI+ cells remains significant, despite a lower p value ($p < 0.05$ in the re-analysis, instead of $p < 0.001$ in our original submission; Figure 2). The proportions of BONCAT cells now decrease during the pre-symptom phase, instead of the symptoms phase as initially described. This reanalysis has been reflected throughout the Results section.

4. The author's also did not perform a vehicle control, which is important, even for longitudinal studies.

As DSS was administered through drinking water (and not through gavage), the suitable vehicle control would have been water over the course of 5 days (duration of DSS administration). While we did not perform a vehicle control, we monitored the mice during 2-4 non-consecutive days over the course of a week (7 days) before DSS exposure to establish a baseline for each experimental repeat (3 experiments, 5 mice every time). Given the variability in bacterial physiology and diversity between cages, using the same mice as their own baseline control is a strength of this study. Indeed, due to large inter-individual variation in gut microbial communities and/or to limit cage-effects, the use of several baseline points in individuals as their own controls has been argued for in other studies such as Fukuyama et al. (PLoS Comput Biol, 2017, 13 (8); Jarett et al., Front Vet Sci, 2021, 8; Nguyen et al., Dis Model Mech, 2015, 8 (1)).

5. The author's also did not repeat the experiment. It appears that this experiment was performed only once.

As mentioned in the previous comment, this experimental setup was independently performed 3 times, with 5 mice per cage each time. This is described in Figure 1a and we have clarified this throughout the revised version.

MINOR COMMENTS

1. The term diversity has some very specific meanings and it is used inappropriately in numerous places in the manuscript. For example, in the sentence below (Lines 84-86), since diversity does not mean specific taxa, this is not an appropriate use -

"composition" would be a better in this particular sentence.

"The changes in bacterial diversity are consistent, with decreases in the short-chain fatty acid (SCFA)-producers Clostridiales, and increases in the more oxygen-tolerant and pro-inflammatory Enterobacteriaceae[11,12]."

We thank the reviewer for this comment and have modified "bacterial diversity" to "bacterial composition" where appropriate.

September 11, 2021

Dr. Corinne F Maurice
McGill University
Microbiology & Immunology
Montreal, QC H3G 0B1
Canada

Re: mSystems00507-21R1 (Changes in gut bacterial translation occur before symptom onset and dysbiosis in dextran sodium sulfate-induced murine colitis)

Dear Dr. Corinne F Maurice:

Thank you for submitting your manuscript to mSystems. We have completed our review and I am pleased to inform you that, in principle, we expect to accept it for publication in mSystems. However, acceptance will not be final until you have adequately addressed the reviewer comments, as reviewer #3 raised valid concerns on how data is being presented in the manuscript.

Preparing Revision Guidelines

Sincerely,

Mariana Byndloss

Editor, mSystems

Journals Department
Reviewer comments:

Reviewer #3 (Comments for the Author):

MAJOR COMMENTS

The authors have improved the paper based on all of the reviewers suggestions.

Concerning this reviewers previous comment about the lack of replication, I believe I now understand the experimental design, which was 3 cages of 5 mice were independently examined, resulting in a total of 15 mice for the entire study.

However, since two cages had 5 male mice each, there is only one cage of 5 female mice. Thus, there is no repeat for the female mouse experiment. It is less than ideal to merge all three datasets without having repeated the female mouse experiment. A much better experimental design would have included a repeat of both the male experiment and the female experiment, and if at that point the results were deemed statistically the same between the sexes, then all of the results could be merged. Thus, without further experimentation, this reviewer does not think the authors should present merged data from 15 mice. This reviewer thinks the authors can present the merged results of the two cages of males (n=10) for a given measured parameter, if the results are shown to be statistically the same for that given parameter, and the results from the one cage of the females (n=5). The authors should therefore not present the results of one cage and say it is representative of all 15 mice - which was done in several places in the manuscript and/or figures.

MAJOR COMMENTS

The authors have improved the paper based on all of the reviewers suggestions.

Concerning this reviewer's previous comment about the lack of replication, I believe I now understand the experimental design, which was 3 cages of 5 mice were independently examined, resulting in a total of 15 mice for the entire study.

However, since two cages had 5 male mice each, there is only one cage of 5 female mice. Thus, there is no repeat for the female mouse experiment. It is less than ideal to merge all three datasets without having repeated the female mouse experiment.

A much better experimental design would have included a repeat of both the male experiment and the female experiment, and if at that point the results were deemed statistically the same between the sexes, then all of the results could be merged. Thus, without further experimentation, this reviewer does not think the authors should present merged data from 15 mice.

This reviewer thinks the authors can present the merged results of the two cages of males (n=10) for a given measured parameter, if the results are shown to be statistically the same for that given parameter, and the results from the one cage of the females (n=5). The authors should therefore not present the results of one cage and say it is representative of all 15 mice - which was done in several places in the manuscript and/or figures.

We thank the reviewer for their comments and are glad to have adequately addressed all their previous concerns. With regards to separating the analysis by sex and providing this information in the main figures and analyzes, we respectfully disagree, as our experimental design was not meant to determine sex-specific changes in bacterial physiology in response to DSS-induced colitis. Our rationale for merging the data was that there are known sex-differences in mice response to DSS, with male mice typically responding more severely than female mice to DSS (Bábičková *et al.* Inflammation, 2015.; Wagnerova *et al.* Gastroenterol. Res. Prac., 2017), and we wished to capture some of this variability in our dataset. There is precedence for including both sexes without showing the differences between them, even in unequal experimental designs (Jernberg *et al.* ISME 2007, Le Gall *et al.* J. Proteome Res. 2011, Marcobal *et al.* ISME 2013, Stacy *et al.* Cell 2021). Importantly, we don't believe that repeating our experiments with another cage of 5 female mice would allow for an appropriate sex analysis, as the final n would still be too low for this type of analysis.

Nevertheless, we did reanalyze our data according to sex, as we agree with the reviewer that this is important data to show. Because of the low n, we believe the sex differences in physiological data we report are preliminary, and we have decided to include these data and figures as Supplementary information instead.

We have therefore added the following paragraph in the Results section to highlight this: “As *sex-specific responses have been noted in DSS-induced colitis, a breakdown by sex is in Supplemental Figure 2 [56]. While there are differences in how bacterial physiology changes in response to a DSS-perturbation, we cannot attribute these differences to sex-specific effects due to the low sample sizes within each sex. Further experiments with larger sample sizes would be needed to confirm the differences seen here. Overall, sex accounted for 4.4%*

and 8.76% of the variation in the proportion of PI+ bacteria and BONCAT-labelled bacteria, with no significant effect of sex on the proportion of HNA bacteria over time.” (L269-276)

October 20, 2021

Dr. Corinne F Maurice
McGill University
Microbiology & Immunology
Montreal, QC H3G 0B1
Canada

Re: mSystems00507-21R2 (Changes in gut bacterial translation occur before symptom onset and dysbiosis in dextran sodium sulfate-induced murine colitis)

Dear Dr. Corinne F Maurice:

Your manuscript has been accepted, and I am forwarding it to the ASM Journals Department for publication. For your reference, ASM Journals' address is given below. Before it can be scheduled for publication, your manuscript will be checked by the mSystems senior production editor, Ellie Ghatineh, to make sure that all elements meet the technical requirements for publication. She will contact you if anything needs to be revised before copyediting and production can begin. Otherwise, you will be notified when your proofs are ready to be viewed.

As an open-access publication, mSystems receives no financial support from paid subscriptions and depends on authors' prompt payment of publication fees as soon as their articles are accepted. =

Publication Fees:

We recognize that the video files can become quite large, and so to avoid quality loss ASM suggests sending the video file via <https://www.wetransfer.com/>. When you have a final version of the video and the still ready to share, please send it to Ellie Ghatineh at eghatineh@asmusa.org.

Sincerely,

Mariana Byndloss
Editor, mSystems

Journals Department
Figure S1: Accept
Fig. S4: Accept
Figure S2: Accept
Supplemental Table 1: Accept
Fig S3: Accept
Table S2: Accept